# The Potential of Fluralaner as a Bait Toxicant to Control Pest Yellowjackets in California

**DOI:** 10.3390/insects14040311

**Published:** 2023-03-23

**Authors:** Michael K. Rust, Chow-Yang Lee, Ho Eun Park, Kathleen Campbell, Dong-Hwan Choe, Mary Sorensen, Andrew Sutherland, Casey Hubble, Beatriz Nobua-Behrmann, John Kabashima, Shu-Ping Tseng, Linda Post

**Affiliations:** 1Department of Entomology, University of California, Riverside, CA 92521, USA; 2Placer Mosquito & Vector Control District, Roseville, CA 95678, USA; 3University of California Cooperative Extension, Alameda County, Hayward, CA 94544, USA; 4University of California Cooperative Extension, Orange County, Irvine, CA 92618, USA; 5Department of Entomology, National Taiwan University, Taipei City 10617, Taiwan; 6San Diego Zoo Safari Park, Escondido, CA 92027, USA

**Keywords:** *Vespula pensylvanica*, isoxazoline, fluralaner, hydrogel baits

## Abstract

**Simple Summary:**

The western yellowjacket, Vespula pensylvanica, is an important seasonal pest of recreational and outdoor venues in the western United States. Effective control strategies are limited, and the objective of this study was to determine the potential of the isoxazoline fluralaner as a bait toxicant. A minimum of 27 colonies were detected foraging at an individual monitoring station using microsatellite markers. As colonies disappeared after baiting, new colonies were detected. Minced chicken and hydrogel baits containing 0.022% and 0.045% fluralaner bait significantly reduced the number of foragers.

**Abstract:**

The western yellowjacket, *Vespula pensylvanica* (Saussure), is an important seasonal pest of recreational and outdoor venues in the western United States. Its propensity to scavenge food increases the likelihood of stinging incidences. Control measures are limited to intensive trapping and treating subterranean nests. The only toxicant registered for baiting in the US is esfenvalerate, which is ineffective. The objective of this study was to determine the potential of the isoxazoline fluralaner as a bait toxicant. With microsatellite genotyping, a minimum of 27 different colonies were shown to forage at a single monitoring site. Some colonies disappeared after baiting, and new colonies were detected. The implications for baiting and monitoring are discussed. Minced chicken and hydrogel baits containing 0.022% and 0.045% fluralaner significantly reduced foraging yellowjackets. Several bait applications covering large areas will be necessary to provide long-term control.

## 1. Introduction

In the western United States, the western yellowjacket, *Vespula pensylvanica* (Saussure), is an important pest of recreational areas, theme parks, and outdoor venues, especially those adjacent to native chaparral habitat [1,2,3]. Its propensity to scavenge foods and garbage when native food sources are scarce can lead to the closure of parks and recreational facilities [3,4,5]. In agricultural settings, yellowjackets can disrupt harvesting operations in orchards and grape vineyards [4,6].

Yellowjackets can be a serious problem where many humans aggregate in outdoor venues and recreation facilities. Hornets, bees, and wasps were responsible for 533 human deaths, and about 70% of all deaths were caused by animals from 1991–2001 [7]. Hospital emergency departments reported 404,115 stinging incidents (Hymenoptera: Vespidae) from 2001–2010, representing about 6% of the total insect stings and bites reported [8]. Wasps, hornets, and bees represented > 70% of the arthropod bites or stings reported at emergency rooms from 2010–2015. During this period, there was a 40% increase in costs for medical treatment (>USD1000 per case) [9].

Yellowjacket control strategies are limited to intensive trapping and treatment of ground nest sites. Intensive trapping has yet to provide satisfactory control in most situations. Although perimeter traps around an 8.9-ha orchard trapped about 200,000 *V. pensylvanica* and reduced numbers below pest levels [6], most examples reported high trap counts, but limited to no effects on reducing stinging incidents. Trapping failed to lower the trap counts of *Vespula alascensis* Packard below an annoyance threshold [10]. Eighty traps surrounding a 20.2-ha water park trapped thousands of foragers over the summer, but wasps remained plentiful [11]. Greater than 0.5 million *V. pensylvanica* were trapped in three 0.2-ha sites without any noticeable reduction in foraging activity [10]. In an interceptive trapping program, a peripheral ring of traps surrounding a picnic area caught over 6000 yellowjackets compared with only 208 at the center picnic tables [12]. Intensive trapping around a health spa in southern California collected > 26,000 yellowjackets during the summer, but there were still stinging incidents [2]. Treating subterranean nests can be highly effective [1,5], but nests are frequently off-site and challenging to locate [13]. The only toxicant registered for baiting yellowjackets in the US is esfenvalerate, and it has provided only marginal reductions in foraging activity [2]. Baits consisting of meats or hydrogel and fipronil were effective against *V. pensylvanica* [2,14,15,16] and *V. germanica* [2,17]. However, fipronil is not registered in the US for yellowjacket control.

In the past decade, a novel class of insecticides, the isoxazolines, have been registered as topical and oral therapies to control fleas, ticks, mites, and biting flies in domestic cats and dogs. The isoxazolines have both contact and oral toxicity against a wide range of ectoparasites [18]. The isoxazoline, fluralaner, is an antagonist of γ-aminobutyric acid (GABA) receptors. Its potency is similar to or higher than fipronil to a wide variety of insects belonging to 11 different orders [19,20]. Considering its safety when administrated as an oral or topical drug to cats and dogs, to control fleas and ticks, fluralaner is an ideal candidate to test as a bait toxicant for yellowjacket control.

The activity of fluralaner as a bait toxicant was field tested over two seasons at four different sites. The efficacy of the treatments was monitored with traps containing the attractant heptyl butyrate. Specimens were collected at one site to determine the number of colonies baited and if new colonies appeared after a bait treatment. The implications of the number of yellowjacket colonies detected before and after baiting and the potential of fluralaner as a bait toxicant for yellowjacket control are discussed.

## 2. Methods and Materials

### 2.1. Field Sites

The UC Berkeley Richmond Field Station (RFS) is approximately 9.7 km northwest of the main UC Berkeley campus (37°54′47.57″ N, 122°20′02.93″ W, elev. 28 m). This area is a natural coastal grassland environment (Figure 1 and Figure 2). It consists of 68.8 ha of which 40.5 ha are uplands, and the remaining area is marsh or bay lands.

Irvine Regional Park (IRP, 33°47′46.82″ N, 117°45′19.82″ W, elev. 180 m) is a multiple-use park (≈64.7 ha) surrounded by undeveloped wilderness areas composed primarily of a riparian, coastal sage scrub, and oak woodland plant community. The park offers many activities, including picnics, concession stands, horse stables, shady turf areas, a zoo, and a small lake (see Appendix A).

The San Diego Zoo Safari Park (SDZ) is a 728.4 ha animal sanctuary located near Escondido, CA (33°05′50.80″ N, 116°59′44.60″ W, elev. 137 m). It is surrounded by coastal sage scrub and chaparral, which are ideal habitats for *V. pensylvanica*. Several areas within the park have had severe yellowjacket problems.

Silent Valley Recreational Vehicle Camp (SV) is located about 8 km south of Banning, CA in the San Jacinto Mountains (33°50′57.51″ N, 116°51′08.45″ W; elev. 1093 m). The camp is about 186 ha with 850 campsites, and supports multiple activities, including a small lake, swimming pools, restaurant, and general store (see Appendix A). The campground is covered with numerous oaks, and the park is surrounded by native chaparral. There have been sporadic yellowjacket problems over the past decade.

### 2.2. Monitoring

At RFS, SV, and IRP, the UCR-style trap constructed from a 946 mL (32oz.) plastic bleach bottle was used to monitor foraging activity (Figure 3). A hole was drilled in the bottle, and a 9- or 13-mL glass vial with a piece of dental wick was screwed into the hole. The glass vial contained a 7.6-cm piece of dental wick and 8 to 13 mL of heptyl butyrate [10]. Wasps were collected in a solution of antifreeze coolant diluted with water (propylene glycol 70:30 vol:vol, Sierra^®^ Antifreeze/Coolant, Old World Industries, Inc., Northbrook, IL, USA) [2]. The solution was effective in killing and preserving the insects. The traps were hung from a wire under trees and bushes about 0.5–1.5 m off the ground. The traps were installed about 50 m apart on the perimeter of the RFS (23 and 15 traps, Figure 1 and Figure 2, respectively) and IRP (58 traps, Appendix A). At SV, 55 traps were hung about 200 m apart along the perimeter of the property (see Appendix A). The heptyl butyrate vials were replenished as needed.

SDZ was monitored with Rescue Disposable Yellowjacket Traps (Sterling International Inc., Spokane, WA, USA) containing heptyl butyrate. Traps were deployed about 50 m apart around six sites in the park (8–15 traps at each site). Traps were hung under trees and bushes about 0.5–1.5 m off the ground. A solution of propylene glycol coolant with water (1:2 dilution) was added to the collection bag. The bag’s contents were removed, and the excess fluid was drained. The insects were placed into 3.785 L plastic zip lock bags and shipped to UC Riverside where the number and species were counted.

Trapping periods varied from 1 to 3 weeks, depending on the availability of personnel, operational considerations at each site, and COVID-19 restrictions on field research. 

### 2.3. Bait Stations

At RFS, SV, and IRP, baits were applied in UCR-style stations. Each station was constructed from two pieces of pine board about 18 cm × 18 cm × 1.8 cm thick and a piece of 2.54-cm mesh hardware cloth [21]. The hardware cloth was stapled to the edges of the boards to construct a cage, and one side of the cloth was not fastened to the wood, allowing for bait cups to be placed inside the cage and later fastened with a twist tie. The bait stations were hung from a piece of wire and a Perky-Pet^®^ ANT GUARD^®^ (Woodstream Corp., Lititz, PA, USA) to prevent ants from feeding on the baits.

At SDZ, UCR circular bait stations were used. The station was constructed from plexiglass disks (29.2 cm diameter, 5 mm thick), 2.54 cm PVC pipe and hardware cloth (1.27 by 2.54 cm mesh, Figure 4). Three bait stations containing four cups of bait were deployed about 10 m apart. The stations were hung from a bush, tree, or on Shepard’s hooks with a wire and an ant guard to prevent ants from feeding on the bait.

### 2.4. Bait Preparation

Baits prepared with chicken or chicken juice were not attractive to the non-pest species of yellowjackets [22]. The fluralaner concentrations, 0.022 and 0.045%, were selected because fipronil was effective at those concentrations and fluralaner has similar activity against insects as fipronil [20]. The 0.045% bait was prepared by mixing 420 g of minced Swanson’s White Premium Chunk Chicken (Campbell Soup Co., Camden, NJ, USA), 80 mL of chicken juice strained from the Swanson’s chicken, and 0.8 mL fluralaner (Bravecto^®^ 280 mg/mL, Intervet Inc., Madison, WI, USA). The 0.022% bait was prepared with the same recipe and 0.4 mL fluralaner. The mixture was thoroughly mixed and minced. The bait was refrigerated overnight.

Hydrogel bait was prepared according to Choe et. al. [14]. The liquid contents from cans of Swanson’s White Premium Chunk Chicken (Campbell Soup Co., Camden, NJ, USA) were strained through cheesecloth. The chicken juice was then diluted with water (1:1) to make a 500 mL suspension to which 33.3 g of polyacrylamide crystals were added (PAA, Watering Storing Crystals, Miracle-Gro Lawn Products, Inc., Marysville, OH, USA). Then, 0.4 and 0.8 mL fluralaner were added to the juice and crystals to produce 0.022 and 0.045% bait, respectively. The mixture was stirred and left overnight in the refrigerator to allow the crystals to swell (referred to hereafter as conditioning).

Plastic cups and lids (59 mL, 2 oz Soufflé Cups and Lids, First Street, Amerifoods Trading Co., Los Angeles, CA, USA) were weighed and ≈30 g of bait was added to each cup. The cup (cup + lid + bait) was weighed again. The bait was refrigerated and transported to the field on ice packs in a cooler. After the exposure, the bait cups were retrieved, placed on ice packs, returned to the laboratory, and weighed.

### 2.5. Evaporation Controls

The two different bait stations were modified to determine the water loss from baits during baiting. The openings in the hardware cloth were covered with a window screen (1 mm mesh) to exclude yellowjackets. The evaporation station was hung near the bait stations. 

Plastic cups and lids were weighed, and ≈30 g of bait was added to each cup. Then, the entire bait cup (cup + minced chicken + lid or cup + hydrogel + lid) was weighed again. The bait was refrigerated until tested. Bait cups were transported to the field on ice packs in a cooler. 

After exposure, the cups were sealed with the lids, returned to the laboratory, and weighed. The average ratio of the Evaporative Initial Bait weight (EIBw)/Evaporative Final Bait weight (EFBw) was calculated. The following calculation determined the amount of bait or food material taken with corrections for the water loss of the bait remaining at the end of the exposure. The amount of bait taken = Initial Bait weight—[Avg. EIBw/EFBw × (Final Bait weight)].

One station with five cups served as the evaporative control in each trial.

### 2.6. Choice Tests

Choice tests were conducted to determine the acceptability of a series of fluralaner concentrations mixed in minced chicken. At IRP and SV, choice feeding tests were conducted in the UCR circular stations. Baits were deployed on 8 September 2020 and retrieved after 24 and 4 h at IRP and SV, respectively.

A bait containing 0.045% fluralaner was prepared by mixing 0.8 mL fluralaner in 80 mL chicken juice and 420 g of finely minced chicken. The 0.045% bait was diluted with untreated minced chicken to provide 0.022, 0.011, and 0.0055% baits and refrigerated overnight.

Plastic cups and lids were weighed, and ≈30 g of bait was added to each cup. Then, the cup (cup + minced chicken + lid) was weighed again. The baits were transported to the field on ice packs in a cooler.

Three choice arenas were tested at each site. The arenas were placed at three locations in IRP and SV with high yellowjacket activity. One cup of each concentration and an untreated cup of minced chicken were placed in a choice arena. Another arena covered with window screen to prevent yellowjackets from foraging on the bait served as the evaporation control. After testing, the cups were covered, placed on ice packs, returned to the laboratory, and weighed. After adjusting for water loss of the baits, the amount of bait removed was determined.

### 2.7. Efficacy Studies

#### 2.7.1. Field Trials Conducted in 2020

Three transects were monitored at RFS during 2020, with 11 weekly trapping periods, beginning 12 May 2020 and ending 21 October 2020 (Figure 1). Baiting trials were conducted when the number of yellowjackets trapped per trap per day (YJ/T/D) value exceeded 10 [2]. The minimum number of YJ/T/D increased to >10 by mid-August.

The 0.022% fluralaner in minced chicken was refrigerated overnight and shipped on ice packs to Richmond. The bait was deployed during daylight hours on two consecutive days along transect A at trap−1A (Figure 1). Three bait stations, each with five plastic cups filled with bait (mean total mass ≈28 g, mean mass bait ≈25 g), were hung about 1.5 m high and about 20 m apart. An evaporation check station with four cups of bait was hung alongside the central bait station. Bait stations were set out in the morning (10:00 h on 27 August 2020 and 09:00 h on 28 August 2020) and removed before sundown (19:00 h on both days). Bait was stored in a refrigerator overnight between these two baiting events. After day 2, the bait cups were removed, covered, and weighed. The impact of baiting was evaluated by comparing the monitoring traps before and after baiting.

A second bait trial late in the season was conducted on 28 September 2020 and 29 September 2020 with 0.022% fluralaner in minced chicken. This trial took place during a late-season heatwave when daytime temperatures exceeded 30 °C. Baits were deployed at site −4A because the wasp counts along transect A remained above the threshold of 10 YJ/T/D value and were highest at –4A. After day 2, the bait cups were removed, covered, and weighed. 

At SV, a transect over 700 m long was baited on 21 September 2020. Three bait stations with 0.022% fluralaner in minced chicken (north end of the transect) and three stations with 0.045% fluralaner (south end of the transect) were placed about 100 m apart. The untreated control area was about 900 m from the nearest bait station (see Appendix A). After 24 h, the bait cups were removed, covered, and weighed. Monitors were collected every two weeks.

At SDZ, the sites were baited with 0.045% fluralaner in minced chicken on 1 September 2020. After 48 h, the cups were covered, placed on ice, returned to the laboratory, and weighed. The untreated control area was 1100 m away from the baited sites. Monitoring traps were collected every 3 weeks to reduce the cost of labor and traps.

#### 2.7.2. Field Trials Conducted in 2021

At RFS, an additional transect was established to provide greater distance from the baited transect and serve as an additional seasonal check (Figure 2). The new transect was located about 500 m south of transect A, and 800 m southeast of transect C. Transect B was removed for the 2021 season. Monitoring traps were installed along transects A and C on 1 June 2021 and along transect X on 8 June 2021. Monitoring continued for transects A and C until 8 November 2021 and 1 November 2021 for transect X.

The 0.022% polyacrylamide (PAA) bait was deployed 18 August 2021 for 48 h at sites −1A and −2A in three UCR-style bait stations provisioned with 4 bait cups (Figure 2). An evaporation bait station was also provisioned with 4 bait cups and hung alongside the central bait station. After 48 h, all bait cups were removed, covered, and weighed. 

At SDZ, monitoring was initiated on 25 May 2021, and the last traps were collected on 11 November 2021. Two areas were baited on 17 August 2021. Each area was >2100 m away from the untreated control area. The 0.045% PAA bait was conditioned in the refrigerator overnight. The bait cups were transported to the park on ice packs in a cooler. After 24 h, the bait cups were retrieved, returned to the laboratory, and weighed. 

A second baiting was conducted with 0.022% and 0.045% PAA bait on 29 September 2021. The baits were conditioned in the refrigerator overnight and transported to the park on ice packs in a cooler. After 24 h, the bait cups were retrieved, placed on ice packs, returned to the laboratory, and weighed. 

### 2.8. DNA Extraction and Microsatellite Genotyping

DNA was extracted from the thorax of *V. pensylvanica* workers using the DNeasy Blood and Tissue Kit (Qiagen, Hilden, Germany) following the manufacturer’s instructions and stored at −20 °C until used. Five workers from each trap in transect A of Richmond Field Station were randomly selected and scored at eight microsatellite loci: RUFA5, RUFA19, LIST2004, LIST2014, LIST2017, LIST2019, LIST2020, VMA6 [23,24,25]. In some traps with sample sizes smaller than five, fewer than five individuals were scored. The PCR mixtures contained 1–2 µL of template DNA, 0.2 µM of each primer, 7.5 µL PCR Master Mix (Cat# K0171, Thermo Scientific, Waltham, MA, USA), and ddH_2_O (15 µL reactions volume in total). For genotyping, forward primers were labeled with 5′-fluorescent tags (6-FAM or HEX; Integrated DNA Technologies, Coralville IA, USA) for genotyping. PCR conditions consisting of an initial denaturation of 3 min at 95 °C, followed by 15 cycles of 30 s at 95 °C, 30 s at an annealing temperature beginning at 60 °C and decreasing 1 °C each cycle, 30 s at 72 °C, then 25 cycles of 30 s at 95 °C, 30 s at 50 °C, 30 s at 72 °C, followed by a final 7-min extension at 72 °C. The PCR products were analyzed on an ABI-3730 Genetic Analyzer (Applied Biosystems) at the University of Arizona Genomic Analysis and Technology Core Facility (GATC). Microsatellite Analysis Software (available on Thermo Fisher Cloud) was used to visualize and score alleles. 

The degree of relatedness among individual workers was estimated using the maximum likelihood sibship reconstruction method in COLONY ver. 2.0.6.6 [26]. This allowed workers to be grouped into colonies and the minimum number of colonies estimated for the study site. The analysis was carried out with the following settings: female polygamous and male monogamous, outbreeding, dioecious haplodiploid organisms, and genotyping error rates ranged from 0–2.5% per locus. Colony analysis was run five times, using a different random number seed each time, for a maximum likelihood reconstruction of full sibship overall runs.

### 2.9. Data Analyses

The amount of bait removed in the choice studies was analyzed with a one-way analysis of variance and means separated by Tukey’s honest significant difference (HSD). The number of yellowjackets trapped before and after baiting was analyzed with a Wilcoxon’s signed-rank test [27]. 

## 3. Results

### 3.1. Choice Tests

At IRP, there were no significant differences in the amount of fluralaner bait removed (mean ± SD) over 24 h from the cups containing 0.045% (17.3 ± 8.18 g), 0.022% (14.7 ± 8.70 g), 0.011% (17.2 ± 11.04 g), and 0.0055% (15.5 ± 3.58 g) fluralaner, and the untreated chicken (21.8 ± 1.72 g, F_4,10_ = 1.5, *p* = 0.27). About 49% of all the treated bait (194 g) was removed.

At SV, there were no significant differences in the amount of fluralaner bait (mean ± SD) removed from the cups containing 0.045% (8.5 ± 4.76 g), 0.022% (12.0 ± 2.51 g), 0.011% (11.1 ± 4.05 g), and 0.0055% (9.4 ± 3.91 g) fluralaner, and untreated chicken (10.1 ± 3.76 g, F_4,10_ = 0.13, *p* = 0.97). A total of 123.3 g of treated bait was removed. 

### 3.2. Baiting Studies

#### 3.2.1. Field Trials Conducted in 2020

At RFS, all the yellowjackets trapped were *V. pensylvanica*. Transects B and C consistently yielded fewer wasps and were left untreated to serve as seasonal checks. A total of 33 queens and 9266 workers were trapped in 2020.

After compensating for water loss, 172.1 g of bait (≈45.8% of the bait applied) was removed in the first baiting trial. After baiting, nine traps along transect A were monitored weekly to assess foraging populations. When considering only the trap at the baiting site (−1A, Figure 1) and the two traps nearest to the baiting site (0A and −2A), there were 94, 84, and 93% reductions in the YJ/T/D value at days 20, 27, and 34 post-baiting, respectively (Table 1). When all nine traps were considered, there was a significant 85% (*n* = 9, Z = 1.69, *p* = 0.098), 72.2% (*n* = 9, Z = 2.04, *p* = 0.041), and 75.3% (*n* = 9, Z = 2.04, *p* = 0.041) reduction in YJ/T/D value at day 20, 27, and 34 post-baiting, respectively. The YJ/T/D value in the untreated controls increased significantly on day 20 (*n* = 14, Z = 2.24, *p* = 0.025) and day 27 (*n* = 14, Z = 1.96, *p* = 0.05). There was no change in the YJ/T/D value at day 34 (*n* = 13, Z = 1.90, *p* = 0.06).

A second bait trial late in the season, using similar methods as above was conducted with 0.022% fluralaner in minced chicken at site −4A (Figure 1) because the trap counts at sites −2A, −3A, and −4A were > 10 YJ/T/D. After compensating for this water loss, 83.7 g of bait (≈22.3% of the bait applied) was removed. The trap counts at −2A, −3A, and −4A declined by 96 and 84% on days 11 and 20, respectively (Table 2). When considering all nine traps, there were 96% (n = 9, W = 21, p = 0.03) and 94% reductions (n = 9, Z = 2.35, p = 0.019) on days 11 and 20 after baiting, respectively. Along transect B and transect C, there was a statistically significant reduction in the YJ/T/D value on day 20 (n = 14, Z = 2.74, p = 0.003).

At SV, there was a significant 78% (*n* = 9, Z= 2.64, *p* = 0.008) and 90% (*n* = 9, Z= 2.64, *p* = 0.008) reduction in YJ/T/D value at weeks 2 and 4, respectively. The trap counts in the untreated control significantly declined at week 6 (*n* = 8, Z = 2.35, *p* = 0.016), and the monitoring was discontinued. 

At SDZ, the total number of *V. pensylvanica* queens and workers trapped in 2020 was 17 and 41,595, respectively. A few *V. sulphurea* (Saussure) and *V. atropilosa* (Sladan) were collected. 

The yellowjackets at SDZ retrieved 92.4 g of bait in 48 h. The 0.045% fluralaner in minced chicken significantly reduced the YJ/T/D value at days 14 (W = 21, *n* = 6, *p* = 0.03) and 35 (W = 21, *n* = 6, *p* = 0.03) (Table 3). There was a significant decrease in the average YJ/T/D value in the untreated control area on day 14 (*n* = 10, Z = 2.57, *p* = 0.006), but the trap counts were above pre-baiting levels at day 35.

#### 3.2.2. Field Trials Conducted in 2021

At RFS, all the specimens collected were *V. pensylvanica*. A total of 10,532 workers and 2 queens were trapped during 2021. The yellowjackets removed 56.6 g of bait (14.3% of the amount deployed). There was no significant reduction in the YJ/T/D value after baiting. The average YJ/T/D value in the two untreated transects varied over the 35 days, but these differences were not significant (Table 4). 

At SDZ, a few *V. atropilosa* and *V. sulphurea* were trapped on 8 July 2021, but all the remaining 14,914 wasps captured throughout the season were *V. pensylvanica* in 2021. Even though the trap counts were well below the threshold of 10 YJ/T/D, two sites were baited because animals and personnel were still being stung.

When the bait taken was adjusted for water loss, 80.5 and 30.3 g of 0.045% and 0.022% fluralaner were removed, respectively at SDZ (Table 5). There was no significant decline in the YJ/T/D values in the site baited with 0.045% fluralaner bait. On day 42 post-baiting, the site baited with 0.022% fluralaner showed a significant reduction in the YJ/T/D value (W = 21, *n* = 6, *p* = 0.03). The YJ/T/D value did not decline at day 21 in the untreated areas (*n* = 23, Z = 1.79, *p* = 0.07). At day 42, there was a significant reduction in the YJ/T/D value in the untreated area (*n* = 22, Z = 2.64, *p* = 0.007: Table 5).

### 3.3. DNA Extraction and Microsatellite Genotyping

Before baiting, a large number of specimens were selected, but because of operational limitations, only 4% were genotyped. A minimum of 27 colonies were identified along transect A (Table 6, Figure 5). This number declined to 19 when examined 20 days after baiting. Of the 27 colonies detected before baiting, 9 of the same colonies were detected after the first baiting, and only 3 colonies were detected after the second baiting (Figure 6). Sixteen new colonies were detected after the first baiting, and five new colonies were detected after the second baiting. 

Ten different colonies were initially detected along transect B (untreated), which increased to fourteen by the end of the study (Table 6). Along transect C (untreated) 12 different colonies were detected at the beginning, and 11 different colonies were detected at the end of the study.

In 2021, a large number of specimens were collected before baiting. Still, because of operational limitations, only 25 specimens were genotyped from transect A, and a minimum of 8 colonies were identified on 18 August 2021. A minimum of 19 (14 new) and 8 colonies (5 new) were identified at 10 and 73 days after baiting, respectively. Colony 1 represented 48% of the pre-baiting specimens along transect A and was not detected again after baiting.

Twenty-one specimens were genotyped on transect C (untreated) on 18 August 2021. A minimum of 12 colonies were detected. Three of the same colonies were present in transects A and C pre-baiting, but none were present in the 10-day post-baiting sample. Six new colonies were present in transect C 10 days after baiting. Eight specimens were genotyped on transect X (untreated) and a minimum of 7 colonies were identified on 9 August 2021. Six different colonies were detected at 10 days after baiting along transect X.

## 4. Discussion

Different thresholds of foraging activity have been proposed before initiating a baiting program for yellowjackets. Grant et al. [28] recommended that baiting for *V. alascensis* and *V. pensylvanica* should commence when trap counts exceeded the annoyance threshold of 7 YJ/T/D value. Rogers [29] proposed an action threshold when traps exceeded 7 V. alascensis/trap/day. Trap counts below this level were acceptable. MacDonald [30] reported that 6–7 workers flying within a 1 m^2^ area caused severe economic impacts and health threats. Rust et al. [2] adopted a slightly higher threshold of 10 YJ/T/D value. The authors recommended these levels of foraging activity to maximize the efficacy of baits. However, it may be necessary to bait when trap counts are much lower because of complaints, especially in the parks and areas with human activities.

Populations of *V. pensylvanica* above a nuisance threshold appear at 3- to 5-year intervals [30]. The numbers of foragers at IRP and RFS began increasing in 2018. In 2019 and 2020, the YJ/T/D value exceeded 10 at multiple traps at all four sites. In 2021, the numbers declined dramatically. Annual monitoring with traps containing heptyl butyrate is important to detect these cycles and initiate timely control measures.

All concentrations of the fluralaner bait tested were readily taken by foragers and there was no evidence of any repellency. The fluralaner baits were not too fast-acting, allowing foragers to make multiple trips to the bait stations. The fluralaner in minced chicken and PAA gels were palatable and readily accepted by foraging yellowjackets. However, the bait’s longevity in the field was significantly increased when formulated in PAA gels [14,22]. When fluralaner baits were applied early in the season, two applications were necessary to provide satisfactory control. A single application in September or October was sufficient to reduce populations at the end of the season.

The microsatellite genotyping indicated that numerous colonies foraged within an area. Rust et al. [21] reported a minimum of 11 colonies foraging at one site. At RFS, a minimum of 27 colonies were collected at one monitoring site in 2020. A minimum of eight colonies were detected at that site in 2021, reflecting the substantial decline in yellowjacket activity observed with monitoring traps. Native populations of *V. pensylvanica* showed weak nestmate recognition [31], which likely explains the multiple colonies foraging at a single site. The effect of the bait on each of those colonies would vary with the amount of bait retrieved by them. Of the yellowjackets typed pre-baiting in 2021, 48% were from one colony (Colony #1). Being the most prevalent colony at the site, colony #1 would have retrieved a larger amount of bait than other colonies. This would explain colony #1 absence in post-baiting samples. Less prevalent colonies continued to be collected after baiting, suggesting that large areas will need to be baited to ensure that all colonies retrieve enough bait to kill a significant proportion of their foragers and larvae.

Akre et al. [32] reported that 80% of the yellowjackets collected were within 335 m, and 95% were within 548 m of their nest. At RFS, transect A and C were a minimum of 500 m apart. Little information is available about the density of underground nests in native habitats [30]. Still, it is evident from the genotyping that numerous nests must have existed in the area surrounding transect A. Three of the 12 colonies detected in transect C were also found in transect A. This suggests that these nests may be located in between the two transects.

Fluralaner baits routinely provided > 80% reductions in trap counts. In 2021, 0.022% fluralaner bait provided an 11% reduction at day 21 post-bait at RFS, but the genotyping indicated that the number of colonies had dramatically increased after baiting. The colony with the greatest number of foragers before baiting was not detected again at this site. A slight increase or a plateauing of trap counts after baiting may be a consequence of neighboring colonies expanding their foraging territories or a few colonies heavily impacted by the baiting. This would explain the need for additional baiting, especially in situations where the trap counts are increasing in the untreated controls.

The use of traps containing heptyl butyrate as an attractant for *V. pensylvanica* is an important tool in determining when foraging activity is high enough to initiate baiting. Microsatellite genotyping reveals numerous colonies may forage and compete for the bait within a given area. As colonies disappeared after baiting, new colonies entered the baited area. A significant reduction in trap counts indicated that the baiting reduced the populations in the treated area. The plateauing of trap counts suggested that additional baiting was necessary. Large areas should be baited to ensure more colonies are provided with adequate bait. Bait stations should be spread over at least 200 m. Baiting should be repeated several times throughout the summer to ensure that newly invading colonies are adequately treated. *V. pensylvanica* populations in southern California normally decline in late September and October depending on the site’s elevation [2]. However, *V. pensylvanica* produces multiyear nests in California, and foragers are often present late in the fall [33]. Monitoring populations is essential in timing bait applications, especially late in the fall.

## 5. Conclusions

Fluralaner is a promising active ingredient for yellowjacket baits. It is highly palatable and readily accepted when mixed with minced chicken or PAA hydrogel. Numerous colonies of yellowjackets may nest in proximity. Monitoring is essential to determine when baiting should begin and if additional bait treatments are required. Bait stations should be deployed over distances of at least 200 m to ensure all colonies access the bait. Baiting should be repeated throughout the summer to ensure the new colonies foraging in the area are also baited.

## Figures and Tables

**Figure 1 insects-14-00311-f001:**
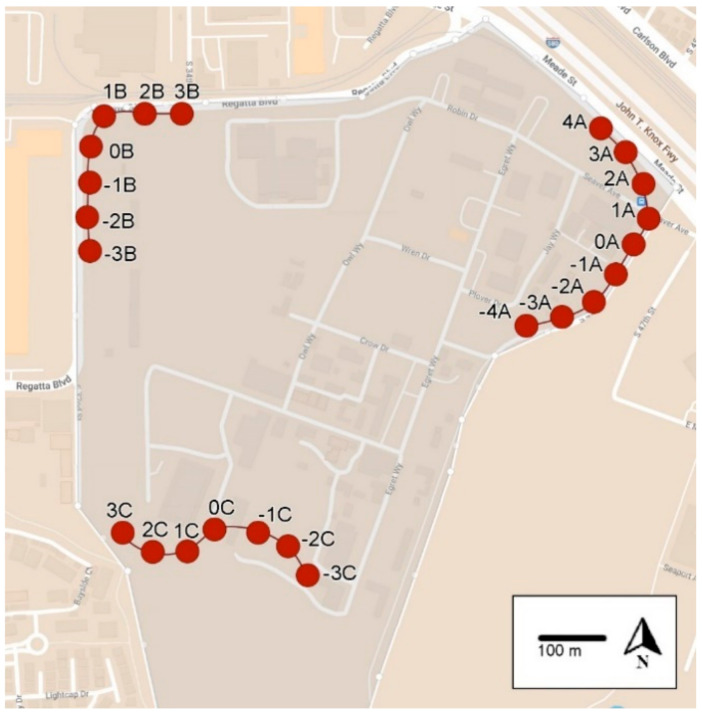
Map of the UC Berkeley Richmond Field Station showing the locations of the three monitoring transects used during 2020. Baits were applied along transect A, and transects B and C were untreated and considered seasonal checks (controls). Red dots indicate the location, and the number of the monitoring trap along transects A, B and C. Minimum distance from transect A to B—600 m; minimum distance from transect A to C—500 m.

**Figure 2 insects-14-00311-f002:**
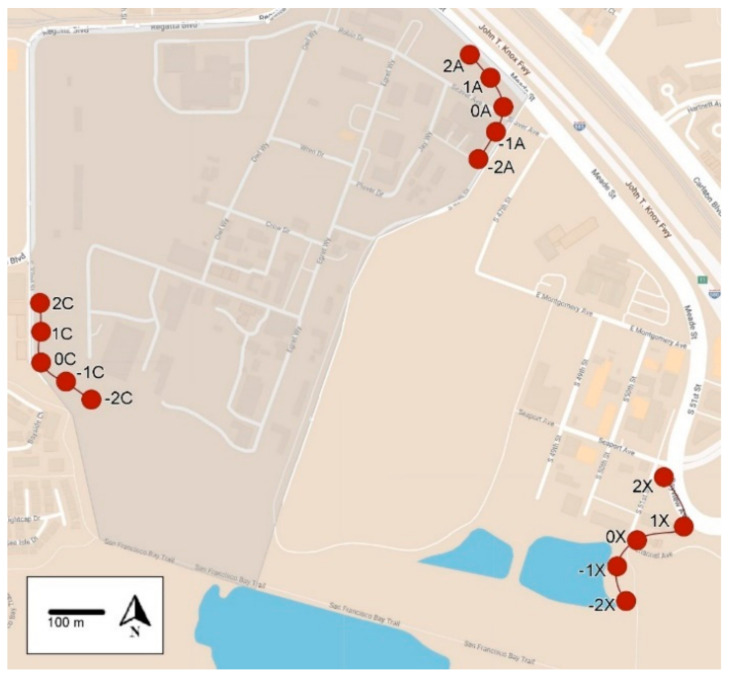
Map of the UC Berkeley Richmond Field Station showing the locations of the three monitoring transects used during 2021. Baits were applied along transect A. Transects C and X were untreated and considered seasonal checks (controls). Red dots indicate the location, and the number of the monitoring trap along transects A, C and X. Minimum distance from transect A to C—800 m; minimum distance from transect A to X—700 m.

**Figure 3 insects-14-00311-f003:**
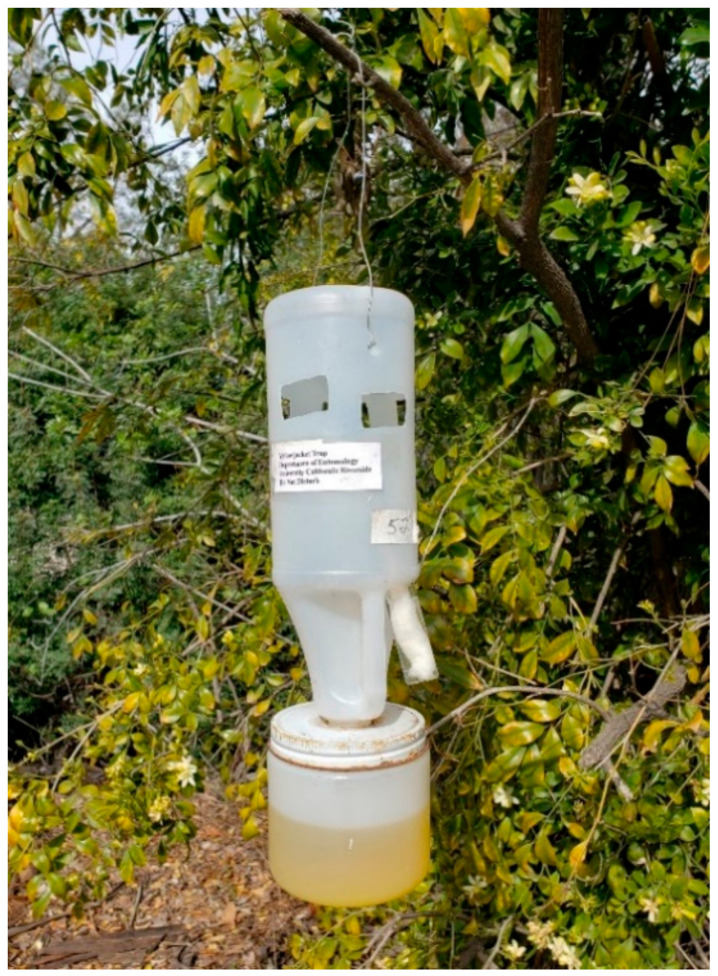
The UCR-style trap with re-useable and interchangeable components.

**Figure 4 insects-14-00311-f004:**
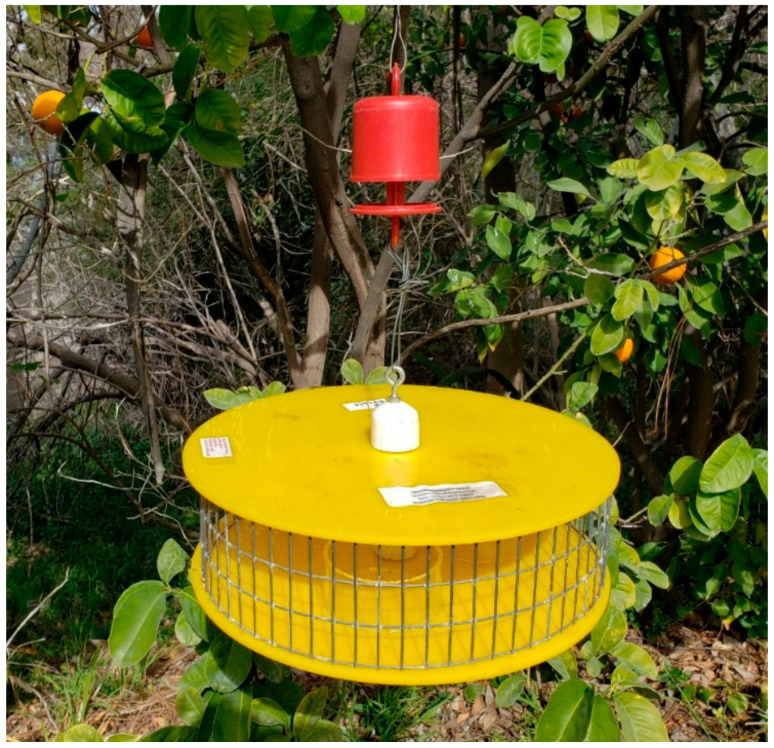
The UCR plastic circular bait stations and ant guard used at the San Diego Zoo Safari Park.

**Figure 5 insects-14-00311-f005:**
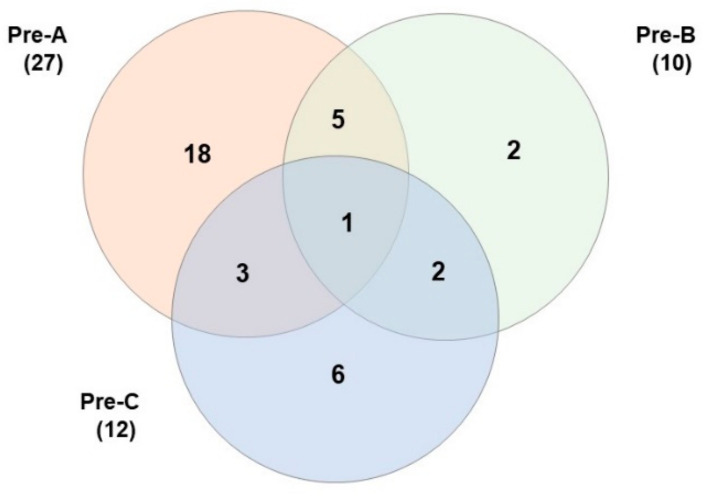
Venn diagram of the number of colonies at transect A, B, and C before baiting in 2020 at RFS.

**Figure 6 insects-14-00311-f006:**
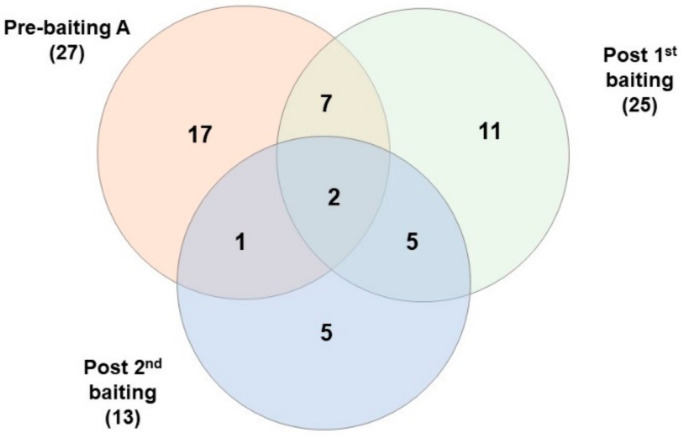
Venn diagram of the number of colonies at transect A before and after baiting in 2020 at RFS.

**Table 1 insects-14-00311-t001:** The average number of yellowjackets trapped per trap per day (YJ/T/D) and percent reduction after baiting with 0.022% fluralaner in minced chicken at the Richmond Field Station ^a^.

Locations	No. Monitoring Traps	Average YJ/T/D (% Reduction) Days after Baiting			
		Pre-Baiting	Day 20	Day 27	Day 34
0A, −1A, −2A	3	26.9	1.5 (94)	3.4 (84)	1.4 (93)
Transect A (all 9 traps)	9	20.2	3.0 (85)	5.6 (72)	5.0 (75)
Transect B (untreated)	7	6.1	16.2 (0)	11.9 (0)	2.3 (62)
Transect C (untreated)	7	4.4	11.9 (0)	9.2 (0)	4.0 (9)

^a^ Bait applied at site −1A on 27 August 2020 and 28 August 2020.

**Table 2 insects-14-00311-t002:** The average number of yellowjackets trapped per trap per day (YJ/T/D) and percent reductions after the second baiting with 0.022% fluralaner in minced chicken at the Richmond Field Station ^a^.

Locations	No. Monitoring Traps	Average YJ/T/D (% Reduction) Days after Baiting		
		Pre-Baiting	Day 11	Day 20
−2A, −3A, −4A	3	27.4	0.5 (96)	1.3 (84)
Transect A (all 9 traps)	9	5.0	0.1 (96)	0.2 (94)
Transect B (untreated)	7	2.3	0.4 (68)	0.2 (88)
Transect C (untreated)	7	4.0	0.8 (60)	0.4 (87)

^a^ Bait was applied at site −4A on 1 October 2020 and 2 October 2020.

**Table 3 insects-14-00311-t003:** The average number of yellowjackets trapped per trap per day (YJ/T/D) and percent reduction after baiting with 0.045% fluralaner in minced chicken at the San Diego Zoo Safari Park ^a^.

Locations	No. Monitoring Traps	Average YJ/T/D (% Reduction)		
		Pre-Baiting	Day 14	Day 35
Treated area	6	14.87	1.88 (87)	1.80 (88)
Untreated area	10	4.61	0.79 (83)	5.06 (0)

^a^ Baited on 1 September 2020.

**Table 4 insects-14-00311-t004:** The average number of yellowjackets trapped per trap per day (YJ/T/D) and percent reduction after baiting with 0.022% fluralaner PAA bait at the Richmond Field Station ^a^.

Locations	Average YJ/T/D (% Reduction) Days after Baiting					
	Pre-Bait	Day 7	Day 14	Day 21	Day 28	Day 35
Transect A (treated)	14.40	9.74 (32)	7.83 (46)	12.83 (11)	10.31 (28)	20.83 (0)
Transect C (untreated)	2.40	5.86 (0)	4.08 (0)	13.5 (0)	8.49 (0)	20.80 (0)
Transect X (untreated)	0.57	0.23 (60)	0.43 (25)	0.54 (5)	0.51 (10)	0.57 (0)

^a^ Five monitoring stations at each transect. Baited 18 August 2021 to 20 August 2021.

**Table 5 insects-14-00311-t005:** The average number of yellowjackets trapped per trap per day (YJ/T/D) and percent reduction after baiting with 0.022% and 0.045% fluralaner in PAA crystals at San Diego Zoo Safari Park ^a^.

Bait	No. Monitoring Traps	Average YJ/T/D (% Reduction)		
		Pre-Baiting	Day 21	Day 42
0.045% fluralaner	12	3.28	1.87 (43)	2.92 (11)
0.022% fluralaner	6	5.86	1.83 (69)	0.84 (86)
Untreated	22	0.94	0.76 (19)	0.38 (60)

^a^ Baited on 29 September 2021 to 30 September 2021 (2nd baiting).

**Table 6 insects-14-00311-t006:** Analysis of yellowjackets at transects A–C with an estimate of the minimum number of different colonies in 2020 at Richmond Field Station.

Transect ^a^	Time of Sample Collection	No. Specimens Collected	No. Specimens Genotyped	Proportion of Genotyped Individuals	Minimum No. Colonies Detected
A	Pre-baiting	1273	45	0.04	27
	20 days after 1st baiting	630	37	0.06	19
	34 days after 1st baiting	314	33	0.11	15
	11 days after 2nd baiting	13	10	0.77	8
	20 days after 2nd baiting	19	11	0.58	7
B	Pre-Baiting	298	34	0.11	10
	20 days after 1st baiting	792	30	0.04	9
	11 days after 2nd baiting	36	29	0.81	14
C	Pre-Baiting	214	26	0.12	12
	20 days after 1st baiting	585	30	0.05	11
	11 days after 2nd baiting	79	24	0.30	11

^a^ Transect A—9 sites sampled, transect B and C—7 sites sampled.

## Data Availability

The data presented in this study are available on request from the corresponding author.

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
