# Peer review of "The Potential of Fluralaner as a Bait Toxicant to Control Pest Yellowjackets in California"

_insects, 2023, doi:10.3390/insects14040311_

Round 1

Reviewer 1 Report

Work into the management of yellowjacket is sorely needed, and this MS makes progress toward development of a commercialized bait that will undoubtedly help control a stinging pest quite capable of rendering stung individuals in the hospital with life threatening reactions to stings.

The authors state that fluralaner is much like fipronil. Is there evidence the yellowjackets are dying from ingestion of the bait, or might it be killing them by contact? Evidence for either/both?

Any evidence of differences in toxicity between the two bait formulations?

L13, italicize scientific name (hereafter I’ll refer to this as ISN because the need appears throughout the MS)

L20, ISN

L34, ISN

L37, delete expensive

L42, further define vespid, maybe …404,115 stinging incidents (Hymenoptera: Vespidae)…

L42 and elsewhere, should incidences be spelled incidents?

L42, delete italics “from 2001 to 2010”

L48, Insert Yellowjacket before Control, and make it a small c

L48, replace treating with treatment of

L50, ISN

L52, incident or incidences

L53, ISN

L54, delete were still and replace with remained

L55, ISN

L60, incident or incidences

L62, insert yellowjackets after baiting; insert comma after esfenvalerate; insert only after provided

L64, ISN

L73, insert for yellowjacket control after toxicant

Figs 1 and 2, Page 3. Why is some of the captioning in both figures italicized? In both figures, what was the distance between and among the transects. Can this be easily put into the caption. I see is interspersed in the MS, but it would be helpful in the captions.

L105, insert yellowjacket after severe

L143. What are the units, after 29.2? millimeters? Centimeters?

L190, replace They were placed out with Baits were deployed

L192, Begin sentence with A bait containing 0.09% fluralaner was prepared…

L216 to 219, it appears that there was only one set up baits, correct, and the bait was deployed and recovered on back to back days, then returned to the lab and weighed. This is not clear.

L190, replace placed out with deployed

L192, Delete The, and start sentence with A bait containing 0.09% fluralaner…

L238, insert the before baited; insert (Figure 2) after check.

L243, define PAA at first mention and then global

L244, insert (Figure 2) after cups

L249, define PPA at first mention and then global

L261, insert period after ]

L301, replace were with was

L306, close space between 9 and %

L309, replace significant increased with increased significantly; day 23 should read day 27

L310, day 30 should read day 34

L317, (Fig. 2) should read (Fig. 1).

L319-323, delete italics

Table 2, footnote of 10/1/2020 to 10/2/2020 does not match the text on line 220 of 9/28 and 9/29; delete italics

L329, delete italics of A at beginning of sentcence

L330, add (data not shown) after respectively.

L333-335, delete italics

Table 3, 0.05% should read 0.045%

Table 4, 0.025% should read 0.022%; bottom right in table move (0.0) up

L361, 0.05% should read 0.045%

Table 5 footnote, insert 8/17/2021 (1st baiting)

Table 6, delete all italics

L384, delete italics

L387, delete #

Are Figs 5 and 6 the same? The numbers in both Venns are the same.

L394-399, looks like font size is smaller than rest of text. Make same.

L407-409 and 416 , ISN

L419-420, delete italics

L422, replace repellency with feeding deterrency

Author Response

Work into the management of yellowjacket is sorely needed, and this MS makes progress toward development of a commercialized bait that will undoubtedly help control a stinging pest quite capable of rendering stung individuals in the hospital with life threatening reactions to stings.

The authors state that fluralaner is much like fipronil. Is there evidence the yellowjackets are dying from ingestion of the bait, or might it be killing them by contact? Evidence for either/both?

Fipronil has contact activity against the foragers and that this can be transferred within the colonies (Rust 2010). The foragers typically spend several minutes cutting and manipulating the chicken or hydrogel baits with their mandibles and legs. The piece of bait is carried back to the nest and fed to the larvae. It is possible that foragers might swallow some liquid in the process, but I would guess that contact activity is responsible for killing foragers.

Any evidence of differences in toxicity between the two bait formulations?

Not currently. If foragers were collected after retrieving the baits and held in cages, it might be possible to determine if there were differences. It might also help to have technical fluralaner, but it is expensive.

L13, italicize scientific name (hereafter I’ll refer to this as ISN because the need appears throughout the MS) Done throughout MS

L37, delete expensive Done

L42, further define vespid, maybe …404,115 stinging incidents (Hymenoptera: Vespidae)…Done

L42 and elsewhere, should incidences be spelled incidents? Done

L42, delete italics “from 2001 to 2010” Done

L48, Insert Yellowjacket before Control, and make it a small c Done

L48, replace treating with treatment of Done

L54, delete were still and replace with remained Done

L62, insert yellowjackets after baiting; insert comma after esfenvalerate; insert only after provided Done

L73, insert for yellowjacket control after toxicant Done

Figs 1 and 2, Page 3. Why is some of the captioning in both figures italicized? In both figures, what was the distance between and among the transects. Can this be easily put into the caption. I see is interspersed in the MS, but it would be helpful in the captions. Done

L105, insert yellowjacket after severe Done

L143. What are the units, after 29.2? millimeters? Centimeters

L190, replace They were placed out with Baits were deployed Done

L192, Begin sentence with A bait containing 0.09% fluralaner was prepared… Done

L216 to 219, it appears that there was only one set up baits, correct, and the bait was deployed and recovered on back to back days, then returned to the lab and weighed. This is not clear. Three bait stations were deployed.

L190, replace placed out with deployed Done

L192, Delete The, and start sentence with A bait containing 0.09% fluralaner… Done

L238, insert the before baited; insert (Figure 2) after check. Done

L243, define PAA at first mention  Done and then global Not sure what you mean by gobal?

L244, insert (Figure 2) after cups Done

L249, define PPA at first mention and then global It should have read PAA.

L261, insert period after ] Done

L301, replace were with was Done

L306, close space between 9 and % Done

L309, replace significant increased with increased significantly; day 23 should read day 27 Done

L310, day 30 should read day 34 Done

L317, (Fig. 2) should read (Fig. 1). Done

L319-323, delete italics Done

Table 2, footnote of 10/1/2020 to 10/2/2020 does not match the text on line 220 of 9/28 and 9/29; delete italics 1 October 2020 to 2 October 2020 is the correct date.

L329, delete italics of A at beginning of sentcence Done

L330, add (data not shown) after respectively. Done

L333-335, delete italics Done

Table 3, 0.05% should read 0.045% Done

Table 4, 0.025% should read 0.022%; bottom right in table move (0.0) up Done

L361, 0.05% should read 0.045% Done

Table 5 footnote, insert 8/17/2021 (1st baiting) Not the correct date. First baiting at another site was with another isoxazoline compound not reported in this paper.

Table 6, delete all italics Done

L384, delete italics Done

L387, delete # Done

Are Figs 5 and 6 the same? The numbers in both Venns are the same. Yes. The wrong jpg was inserted for Figure 6. It has been corrected.

L394-399, looks like font size is smaller than rest of text. Make same. Done

L407-409 and 416 , ISN Done

L419-420, delete italics Done

L422, replace repellency with feeding deterrency I didn’t change because the foragers do not actually feed on the baits. They cut and carry pieces of the baits to feed to the larvae. The foragers are probably killed by contact activity of the insecticide in the bait.

Reviewer 2 Report

Rust et al. present the results of a series of trials on the delivery of fluralaner in bait stations for the control of yellowjackets (V. pensylvanica). A series of trials was performed in 2020 and 2021, backed up with evaporation and toxicant palatability studies. The results will be of interest to pest control practitioners working on vespid control.
I have some general comments and a series of specific numbered comments written on a scanned copy of the manuscript.
General comments.
(i) The clarity of the text needs to be improved in places, particularly with regard to the numbers and spatial arrangement of bait stations and monitoring traps.
(ii) There is a clear positive correlation between the number of insects tested and the number of colonies estimated by PCR genotyping. This means that the ability to draw firm conclusions regarding the number of colonies in the vicinity of a particular site is clearly limited. The authors might know of a metric that takes sample size into account for colony number estimation (I do not).
(iii) Some of the figures can probably be eliminated or modified in light of point (ii) on sample size effects (fig 5 &6?) or reduced in size (figs 3 & 4). An additional figure might be included if this improves understanding of the spatial arrangement of traps relative to bait stations.
Numbered points written on scanned manuscript.
1. Does Insects have an article type "Project report"? I could not find this on the website.
2. The manuscript has numerous formatting issues – some text is grey, or written in italics. This needs to be checked before resubmission.
3. I was looking for information on distances between traps and the number of traps per site here.
4. I was looking for information on trap replication here.
5. Insects is an international journal, but I think that the US is the only country that uses the month/day/year code. To avoid confusion for the rest of the world using the day/month/yr format, I would suggest changing dates to day and month written out (e.g., July 18, or 18 July) rather than the m/d/y format.  This needs changing in the entire manuscript.
6. I did not understand the "treatment threshold" concept. Do you mean that trials were performed only at sites at which YT/T/D values exceeded 10?
7. What does high temperature mean? Is this the maximum daily temperature? Is this an average for the experimental period? Or maybe just say that daily temperatures exceeded 30 °C?
8a. Were these two trials "monitored"?
8b. What is a monitor?  You probably need to explain that the impact of baits was evaluated using monitor traps or treatment evaluation traps.
8c. Would a figure help to explain trap arrangements?
9. Please provide SE or SD values for all mean values in manuscript.
10a. Given that percentage reduction values are based on small numbers of traps, I suggest that percentages should be given to whole numbers rather than to a decimal place.  This would also improve the readability of the results.
10b. What test was applied that generated Z statistics? Not mentioned in the Methods.
11. Were these species excluded from the analyses?
12. Please check the treatments described in the title/text and the tables. They differ.
13. I may be wrong, but I don't think that Table 6 shows number of SAMPLES collected, but it may show the number of individuals collected.
14. Again, this should be individuals, not samples, correct?
15. You obviously have a significant positive correlation between number of individuals analyzed and number of colonies detected (Pearson's P = 0.003 in my estimation – I plotted a small graph for clarity). This has a major effect on your conclusions and ability to draw inferences. You need to make this clear.
16. Are the Venn diagrams in Figs 5 & 6 really much use if they are biased by sample size differences? I would suggest eliminating them, or moving them to suppl. material with appropriate caveats regarding the accuracy of the values.
17 ...enough bait for what?
18. Popular culture has it that it's always sunny in California, but I'm guessing that weather conditions had a significant effect on YJ/T/D values. Did they?
19. Do you have a basis for the immigration of YJ colonies? I would have thought that reproduction among local colonies would be a more parsimonious explanation.
20. Why is monitoring in late fall of particular importance if these colonies are likely to die out anyway?
21. Larger than what?
22. Is there evidence for invasion rather than local reproduction?

Other issues.
The author of affiliation #2 is a commercial company employee. Is there a commercial interest here that needs to be stated, from an ethical standpoint?
Minor formatting issues in the references. URLs are usually followed by a site access date.
Figs 3 & 4 are very large in my opinion and could be combined into a single figure (fig 3a, 3b?).

Author Response

Rust et al. present the results of a series of trials on the delivery of fluralaner in bait stations for the control of yellowjackets (V. pensylvanica). A series of trials was performed in 2020 and 2021, backed up with evaporation and toxicant palatability studies. The results will be of interest to pest control practitioners working on vespid control.I have some general comments and a series of specific numbered comments written on a scanned copy of the manuscript.

General comments.
(i) The clarity of the text needs to be improved in places, particularly with regard to the numbers and spatial arrangement of bait stations and monitoring traps.

The additional information has been added regarding both the monitoring traps and bait stations at all the sites. We also provided 2 additional figures.

(ii) There is a clear positive correlation between the number of insects tested and the number of colonies estimated by PCR genotyping. This means that the ability to draw firm conclusions regarding the number of colonies in the vicinity of a particular site is clearly limited. The authors might know of a metric that takes sample size into account for colony number estimation (I do not).

We don’t know of any metric that takes sample size into account for colony number estimation. Fig. 6 should have included the data regarding colonies before and after baiting (please see the new Fig. 6). The association between the number of samples genotyped and the minimum number of colonies fails to consider the effects of baiting in Transect A or the timeline. The number of yellowjackets collected in traps dramatically declined because of the baiting. In Transect A, initially 27 colonies were detected from 45 yellowjackets sampled. After the first baiting, 37 yellowjackets were genotyped and only 7 of the original colonies were detected and 18 new colonies appeared (new Fig. 6). Five new colonies appeared after the second baiting. Of the 27 colonies initially trapped, only 2 remained after the second baiting.

I think this supports our general conclusion that numerous colonies are foraging and collected at one site. After baiting, most of the colonies disappeared and new colonies were now foraging in that area.

(iii) Some of the figures can probably be eliminated or modified in light of point (ii) on sample size effects (fig 5 &6?) or reduced in size (figs 3 & 4). An additional figure might be included if this improves understanding of the spatial arrangement of traps relative to bait stations.

The figures have been altered in size. I have also included a figure that shows the positioning of traps at Irving Regional Park and Silent Valley (Supplemental files). I can’t show the San Diego Safari Park because the areas are restricted where animal breeding occurs. Additions in the text have also been included to clarify where the traps and bait stations were deployed.

Numbered points written on scanned manuscript.
1. Does Insects have an article type "Project report"? I could not find this on the website.

We didn’t add this.

  1. The manuscript has numerous formatting issues – some text is grey, or written in italics. This needs to be checked before resubmission.

I don’t believe the original submission had these issues, but we have corrected them thorough out the paper.

  1. I was looking for information on distances between traps and the number of traps per site here.

This has been added for each site. Additional maps have been provided.

  1. I was looking for information on trap replication here.

This has been added.

  1. Insects is an international journal, but I think that the US is the only country that uses the month/day/year code. To avoid confusion for the rest of the world using the day/month/yr format, I would suggest changing dates to day and month written out (e.g., July 18, or 18 July) rather than the m/d/y format.  This needs changing in the entire manuscript.  Done.
  2. I did not understand the "treatment threshold" concept. Do you mean that trials were performed only at sites at which YT/T/D values exceeded 10?

Yes. I have tried to clarify this here. There is additional information in the Discussion.

  1. What does high temperature mean? Is this the maximum daily temperature? Is this an average for the experimental period? Or maybe just say that daily temperatures exceeded 30 °C?

Changed to say daily temperatures exceeded 30 C.

8a. Were these two trials "monitored"? Yes. WE clarified this.

8b. What is a monitor?  You probably need to explain that the impact of baits was evaluated using monitor traps or treatment evaluation traps. Done

8c. Would a figure help to explain trap arrangements?

I have added two supplementary figures. They show locations where monitoring traps and bait stations were deployed.

  1. Please provide SE or SD values for all mean values in manuscript.

The figures are the total amount of bait being removed from all three bait stations. These aren’t averages. I can see how this is confusing. It is not the average amount of bait being taken per station that is important. It is the total amount of bait taken.

10a. Given that percentage reduction values are based on small numbers of traps, I suggest that percentages should be given to whole numbers rather than to a decimal place.  This would also improve the readability of the results.

Done. Changes also made in the text.

10b. What test was applied that generated Z statistics? Not mentioned in the Methods.

This z value is from the Wilcoxon signed rank test. (Statistix 10 version). In some cases only the W value was given.

  1. Were these species excluded from the analyses?

Yes. Their numbers were so low and inconsistent it was not possible to analyze them.

  1. Please check the treatments described in the title/text and the tables. They differ.

Done.

  1. I may be wrong, but I don't think that Table 6 shows number of SAMPLES collected, but it may show the number of individuals collected.

I have corrected the table. The numbers do represent specimens or individuals. Transect a had 9 sampling sites and transects B and C had 7. This is also shown in the Figures 1 and 2.

  1. Again, this should be individuals, not samples, correct?

Yes. Changes are made.

  1. You obviously have a significant positive correlation between number of individuals analyzed and number of colonies detected (Pearson's P = 0.003 in my estimation – I plotted a small graph for clarity). This has a major effect on your conclusions and ability to draw inferences. You need to make this clear.

The problem with the association test conducted is that it does consider that transect A was baited twice. I think the Pearson’s association should be conducted on Pre baiting Transects A, Transects B and C data. These are all untreated values. I think the Pearson’s P =0.4736 for this data.

Dr. Tseng sent me the following analysis of some genotyping data from 2020 and 2021. I will try to attach it. This isn’t as definitive. This warrants some additional research.

  1. Are the Venn diagrams in Figs 5 & 6 really much use if they are biased by sample size differences? I would suggest eliminating them, or moving them to suppl. material with appropriate caveats regarding the accuracy of the values.

Fig. 6 has been corrected. I think Fig. 6 clearly shows that some colonies from Transect A no longer appear after baiting. New colonies are detected after each baiting.

17 ...enough bait for what? To kill all the foragers and larvae.

  1. Popular culture has it that it's always sunny in California, but I'm guessing that weather conditions had a significant effect on YJ/T/D values. Did they?

The monitoring is conducted over 1-3 weeks and is unlikely to be greatly affected by weather. The baiting is only conducted when the weather is favorable for the 24-48 hour period. In late September and October weather is more likely to have some impacts, especially at elevated sites like Silent Valley RV Park.

  1. Do you have a basis for the immigration of YJ colonies? I would have thought that reproduction among local colonies would be a more parsimonious explanation.

Little is known about the density of yellowjacket nests. The maintenance personnel at the parks are usually good at finding nests within the parks and treating them or letting us know about them. The yellowjackets are foraging into the parks.  

  1. Why is monitoring in late fall of particular importance if these colonies are likely to die out anyway?

Stinging incidents do occur in the fall and the past Septembers have been unusually warm. V. pensylvanica nests have been known to overwinter. We have baited at the park in San Diego even though the YJ/T/D values were <10. I also added a reference about multiyear nests which are common in southern CA.

  1. Larger than what?

Our three bait stations were typically spread over 30–50 meters. This distance needs to be at least 200 m. I have added this.

  1. Is there evidence for invasion rather than local reproduction?

I think the genotyping shows that some colonies no longer appear after baiting. It also shows that many new colonies appear. Foragers from either overwintering colonies or newly established colonies within a giving area are being trapped in the monitors. We simply don’t know much about the density of these ground nests. Unfortunately, we lost genotyping data in 2021 in a park in southern California because the pre-baiting samples deteriorated.    

Other issues.
The author of affiliation #2 is a commercial company employee. Is there a commercial interest here that needs to be stated, from an ethical standpoint?

Dr. Sorensen is a senior scientist at governmental organization. In California, county governments have vector control departments. They are responsible for monitoring and treating arthropods of medical and urban importance in CA.  

Minor formatting issues in the references. URLs are usually followed by a site access date. Done

Figs 3 & 4 are very large in my opinion and could be combined into a single figure (fig 3a, 3b?).They have been reduced.

Round 2

Reviewer 2 Report

The authors have addressed my concerns. I detected an error in the text in line 406 that could be corrected in the proof stage.